# The AGEs/RAGE Transduction Signaling Prompts IL-8/CXCR1/2-Mediated Interaction between Cancer-Associated Fibroblasts (CAFs) and Breast Cancer Cells

**DOI:** 10.3390/cells11152402

**Published:** 2022-08-04

**Authors:** Maria Francesca Santolla, Marianna Talia, Francesca Cirillo, Domenica Scordamaglia, Salvatore De Rosis, Asia Spinelli, Anna Maria Miglietta, Bruno Nardo, Gianfranco Filippelli, Ernestina Marianna De Francesco, Antonino Belfiore, Rosamaria Lappano, Marcello Maggiolini

**Affiliations:** 1Department of Pharmacy, Health and Nutritional Sciences, University of Calabria, 87036 Rende, Italy; 2Breast and General Surgery Unit, Regional Hospital Cosenza, 87100 Cosenza, Italy; 3Oncology Department, Hospital of Paola, 87100 Cosenza, Italy; 4Endocrinology, Department of Clinical and Experimental Medicine, University of Catania, Garibaldi-Nesima Hospital, 95122 Catania, Italy

**Keywords:** cancer-associated fibroblasts, AGEs, RAGE, IL-8, breast cancer

## Abstract

Advanced glycation end products (AGEs) and the cognate receptor, named RAGE, are involved in metabolic disorders characterized by hyperglycemia, type 2 diabetes mellitus (T2DM) and obesity. Moreover, the AGEs/RAGE transduction pathway prompts a dysfunctional interaction between breast cancer cells and tumor stroma toward the acquisition of malignant features. However, the action of the AGEs/RAGE axis in the main players of the tumor microenvironment, named breast cancer-associated fibroblasts (CAFs), remains to be fully explored. In the present study, by chemokine array, we first assessed that interleukin-8 (IL-8) is the most up-regulated pro-inflammatory chemokine upon AGEs/RAGE activation in primary CAFs, obtained from breast tumors. Thereafter, we ascertained that the AGEs/RAGE signaling promotes a network cascade in CAFs, leading to the c-Fos-dependent regulation of IL-8. Next, using a conditioned medium from AGEs-exposed CAFs, we determined that IL-8/CXCR1/2 paracrine activation induces the acquisition of migratory and invasive features in MDA-MB-231 breast cancer cells. Altogether, our data provide new insights on the involvement of IL-8 in the AGEs/RAGE transduction pathway among the intricate connections linking breast cancer cells to the surrounding stroma. Hence, our findings may pave the way for further investigations to define the role of IL-8 as useful target for the better management of breast cancer patients exhibiting metabolic disorders.

## 1. Introduction

Breast cancer is the most commonly diagnosed type of tumor and the second leading cause of cancer-related death among women worldwide [1]. The functional interactions between cancer cells and the surrounding microenvironment have been involved in the metastatic evolution, which is a main determinant of breast cancer mortality [2,3]. Therefore, a better understanding of the mechanisms hidden behind breast cancer metastasis is urgently needed, to disclose new effective therapeutic strategies. In this context, a growing body of evidence has suggested a relationship between metabolic disorders, such as obesity, hyperglycemia and type 2 diabetes mellitus (T2DM) and both increased breast cancer risk and tumor-related mortality [4,5,6,7,8,9,10,11]. Previous studies, nicely supporting these observations, have reported that the aforementioned chronic pathological conditions are characterized by the release of various factors, prompting an inflammatory landscape toward breast cancer development [12,13]. For instance, an increase in the advanced glycation end-products (AGEs) has been observed under inflammatory conditions in patients affected by metabolic syndromes [14,15,16]. AGEs are heterogeneous and harmful compounds derived from an irreversible nonenzymatic reaction between the carbonyl group of a reducing sugar and a free amino group of proteins, lipids or nucleic acids [17,18]. AGEs can be generated endogenously or provided by exogenous sources in physiological and pathological conditions [14,19,20]. The biological effects of AGEs are mainly mediated by the binding to the receptor for advanced glycation end-products (RAGE), which is a transmembrane protein belonging to the immunoglobulin superfamily expressed in both normal and cancer cells [21,22,23]. For instance, in different types of tumors including breast cancer, RAGE activation may lead to proliferative, angiogenic, migratory and invasive responses associated with malignant features and a worse prognosis [24,25,26]. By orchestrating a feed-forward autocrine and/or paracrine loop that entails both the breast cancer cells and the surrounding stroma, RAGE acts as an important mediator, linking chronic inflammation to cancer progression [23,27,28,29].

Pro-inflammatory mediators, such as IL-8, play an important role in stimulating pro-metastatic effects in diverse types of tumors, including breast cancer [30,31,32,33,34,35]. IL-8 is a small soluble protein that belongs to the CXC chemokine family [36]. Initially, IL-8 was recognized as a modulator of neutrophil activity and a chemotactic agent released by monocytes and macrophages upon diverse stimuli [37,38]. Nowadays, IL-8 is primarily acknowledged as a potent mediator of tumor progression, due to its ability to induce angiogenesis, survival, invasion and metastasis in both an autocrine and paracrine manner [39,40]. As it concerns breast cancer, high IL-8 expression was assessed in breast tumors in relation to normal breast tissues along with its ability to stimulate the malignant progression [41]. Nicely fitting with these data, the inhibition of IL-8 blunted the invasiveness of breast cancer cells either in vitro or in vivo model systems [42,43,44,45,46,47].

Compelling evidence has demonstrated that the main components of the tumor stroma, named cancer-associated fibroblasts (CAFs), play an active role in breast cancer through the secretion of cytokines, chemokines, growth factors and other mediators within the tumor microenvironment [48,49,50,51,52,53]. This dynamic network may prompt the failure of therapeutics, metastatic features and poor clinical outcomes [54,55,56,57]. Of note, recent studies have also revealed that CAFs may facilitate the establishment of a supportive microenvironmental niche for the recruitment and growth of disseminated tumor-initiating cells [58,59].

On the basis of these observations, we aimed to provide novel insights into the action of the AGEs/RAGE axis within the tumor microenvironment toward the acquisition of invasive and aggressive features of breast cancer cells.

## 2. Materials and Methods

### 2.1. Bioinformatics Analyses 

The gene expression levels and clinical information were retrieved from 17 integrated Affymetrix gene expression datasets, as previously described [60]. In brief, the Raw.cel files from 17 Affymetrix U133A/plus two gene expression datasets of primary breast tumors were retrieved from NCBI GEO (GSE12276, GSE21653, GSE3744, GSE5460, GSE2109, GSE1561, GSE17907, GSE2990, GSE7390, GSE11121, GSE16716, GSE2034, GSE1456, GSE6532, GSE3494), summarized with Ensembl alternative CDF [61], and then normalized with RMA [62], before their integration using ComBat [63] to eliminate dataset-specific bias [64]. A comprehensive survival analysis was performed, using the Affymetrix gene expression data of CXCR1 and CXCR2 along with the relapse-free survival (RFS) information of basal breast-cancer patients, which were filtered for missing values. The survivALL package was employed to examine the Cox proportional hazards for all of the possible points-of-separation (low–high cut-points), selecting the cut-point with the lowest *p*-value [65] and separating the patients into high (*n* = 254) and low (*n* = 38) CXCR1/2 expression levels. The Kaplan–Meier survival curve was generated using the survival and the survminer packages. All of the bioinformatics analyses were carried out using R Studio (Integrated Development Environment for R. RStudio, PBC, Boston, MA, USA). The box plots were performed with the tidyverse package and the related statistical analysis was performed using the Wilcoxon test. *p*-values < 0.05 were considered statistically significant.

### 2.2. Reagents

We purchased AGEs-BSA (AGEs) from Abcam (DBA, Milan, Italy); the FPS-ZM1, BSA and N-acetyl-L-cysteine (NAC) from Merck Life Science (Milan, Italy). The Trametinib, Alpelisib and Reparixin were obtained from MedChemExpress (DBA, Milan, Italy). The anti-IL-8 neutralizing antibody (MAB208) was acquired from R&D Systems (Bio-Techne, Milan, Italy). All of the compounds were solubilized in dimethyl sulfoxide (DMSO), except for AGEs-BSA and BSA that were dissolved in phosphate-buffered saline (PBS), and NAC that was solubilized in water. 

### 2.3. Cell Cultures

The cancer-associated fibroblasts (CAFs) were isolated, cultured and characterized as previously described [28,66] from 10 invasive ductal breast carcinomas and pooled for the subsequent studies. Briefly, the specimens were cut into small pieces (1–2 mm diameter), placed in a digestion solution (400 IU collagenase, 100 IU hyaluronidase and 10% FBS, containing antibiotics and antimycotics solution) and incubated overnight at 37 °C. The cells were then separated by differential centrifugation at 90× *g* for 2 min. The supernatant containing fibroblasts was centrifuged at 485× *g* for 8 min, the pellet obtained was suspended in fibroblasts’ growth medium (Medium 199 and Ham’s F12 mixed 1:1 and supplemented with 10% FBS and 1% penicillin/streptomycin) (Thermo Fisher Scientific, Monza, Italy) and cultured at 37 °C, 5% CO_2_. The CAFs were then expanded into two 15-cm Petri dishes and stored as the cells passaged for two to three population doublings within a total of 7 to 10 days after tissue dissociation. The primary cell cultures of the breast fibroblasts were characterized by immunofluorescence. In particular, the cells were incubated with human anti-vimentin (V9, sc-6260) and human anti-cytokeratin 14 (LL001 sc-53,253), obtained from Santa Cruz Biotechnology (DBA, Milan, Italy) (data not shown). To characterize the fibroblasts’ activation, we used anti-fibroblast activated protein α (FAPα) antibody (H-56; Santa Cruz Biotechnology, DBA, Milan, Italy) (data not shown). The MDA-MB-231 cells obtained from the ATCC cells (Manassas, VA, USA) were cultured in DMEM/F12 (Dulbecco’s modified Eagle’s medium) with phenol red, 5% fetal bovine serum (FBS) and 1% penicillin/streptomycin (Thermo Fisher Scientific, Monza, Italy). The cells were used less than 6 months after resuscitation and routinely tested for mycoplasma. All of the cell lines were grown in a 37 °C incubator with 5% CO_2_. 

### 2.4. Gene Expression Studies

The total RNA was extracted and cDNA was obtained by reverse transcription, as previously reported [67]. The expression of the selected genes was analyzed by real-time PCR using platform Quant Studio7 Flex Real-Time PCR System (Thermo Fisher Scientific, Monza, Italy). The following primers were used: 5′-AAGCCACCCCACTTCTCTCTAA-3′ (ACTB Fwd.) and 5′-CACCTCCCCTGTGTGGACTT-3′ (ACTB Rev); 5′-TGTGGGTCTGTTGTAGGGTT-3′ (IL-8 Fwd.), 5′-TCGGATATTCTCTTGGCCCT-3′ (IL-8 Rev); 5′-CGAGCCCTTTGATGACTTCCT-3′ (c-Fos Fwd.) and 5′-GGAGCGGGCTGTCTCAGA-3′ (c-Fos Rev); 5′-ACAGTGGCCACCTACAAAGG-3′ (N-cadherin Fwd.) and 5′-CCGAGATGGGGTTGATAATG-3′(N-cadherin Rev); 5′-TCCGCACATTCGAGCAAAGA-3′(vimentin Fwd.) and 5′-ATTCAAGTCTCAGCGGGCTC-3′(vimentin Rev); 5′-CAGTGGGAGAACCTCGAGAAG-3′ (fibronectin Fwd.) and 5′-TCCCTCGGAACATCAGAAAC-3′ (fibronectin Rev). The primers were designed using Applied Biosystems Primer Express 2.0 software. The assays were performed in triplicate and the results were normalized with control mRNA levels of beta-actin (ACTB) and relative mRNA levels were calculated, using the comparative cycle threshold (Ct) method (^ΔΔ^Ct).

The PCR arrays were carried out using a TaqMan™ Human Chemokines Array (Thermo Fisher Scientific, Monza, Italy), according to the manufacturer’s instructions. The amplification reaction and the subsequent analysis were performed, using the platform Quant Studio7 Flex Real-Time PCR System (Thermo Fisher Scientific, Monza, Italy). The heatmaps were drawn on the log2 fold changes of gene expression using the pheatmap package.

### 2.5. Luciferase Assays and Gene Silencing Experiments

The IL-8 promoter luciferase construct was a kind gift from Prof. Richard O.C. Oreffo, Institute of Development Sciences, University of Southampton, Southampton (UK) [68]. The expression vector for the Renilla luciferase pRL-TK (Promega, Milan, Italy) was used as an internal transfection control. The transfections were performed using X-treme GENE 9 DNA Transfection Reagent, according to the manufacturer’s instructions (Merck Life Science, Milan, Italy), with a mixture containing 0.5 µg of IL-8 reporter plasmid and 5 ng of pRL-TK. After 8 h, the cells were treated for 18 h, as indicated. Then, the luciferase activity was normalized to the internal transfection control provided by the Renilla luciferase activity. The normalized relative light-unit values obtained from the cells treated with the vehicle were set as one-fold induction upon which the activity induced by the treatments was calculated. For knocking down the RAGE expression, the cells were transiently transfected with Lipofectamine RNAiMAX (Thermo Fisher Scientific, Monza, Italy), using a pool of three unique 27mer siRNA duplexes for RAGE (AGER, #SR319295) targeting-sequences (siRAGE) (10 nM) or a non-targeting scramble control (10 nM) for 24 h prior to the treatments (OriGene Technologies, DBA, Milan, Italy). The plasmid DN/c-Fos, which contains the sequence for a mutant form of the c-Fos protein that heterodimerizes with the c-Fos dimerization partners but does not allow DNA binding, was generously gifted to us by Dr C. Vinson (NIH, Bethesda, MD, USA).

### 2.6. Western Blot Analysis

Western blotting analyses were carried out, as previously described [69]. The primary antibodies used are as follow: IL-8 (27095-1-AP) (Proteintech, DBA, Milan, Italy); p-AKT-(Ser473) (D9E) and RAGE (42544S) (Cell Signalling Technology, Euroclone, Milan, Italy); β-actin (AC-15), p-ERK1/2 (E-4), ERK2 (C-14), AKT/1/2/3 (H-136) and c-Fos (E-8) (Santa Cruz Biotechnology, DBA, Milan, Italy). The proteins were detected by horseradish peroxidase-linked secondary antibodies (Bio-Rad, Milan, Italy) and revealed, using the chemiluminescent substrate Clarity Western ECL Substrate (Bio-Rad, Milan, Italy).

### 2.7. DCFDA Fluorescence Measurement of ROS

The intracellular ROS production was evaluated, using the non-fluorescent 2′,7′-dichlorofluorescin diacetate (DCFDA) probe, which becomes highly fluorescent when reacting with ROS. Briefly, the cells were treated as indicated, and then washed with PBS and incubated at 37 °C for 30 min with 10 μM DCFDA (Sigma-Aldrich, Milan, Italy). Next, the cells were washed with PBS, and the fluorescent intensity of DCF was measured with excitation at 485 nm and emission at 530 nm, using the software Gen5 in Synergy H1 Hybrid Multi-Mode Microplate Reader (BioTek, AHSI, Milan Italy).

### 2.8. Immunofluorescence Microscopy

Fifty percent of the confluent-cultured cells grown on coverslips were serum deprived for 24 h and then treated for 6 h, as indicated. Thereafter, the cells were fixed with 4% paraformaldehyde (PFA) diluted in PBS for 10 min at room temperature. Then, after a brief rinsing with PBS, the slides were incubated for 90 min with Phalloidin-Fluorescent 488 Conjugate (Santa Cruz Biotechnology, DBA, Milan, Italy). Next, the slides were extensively washed with PBS and probed with 4, 6-diamidino-2-phenylindole dihydrochloride (DAPI) (1:1000; Sigma-Aldrich, Milan, Italy). The images were acquired using the Cytation 3 Cell Imaging Multimode reader (BioTek, AHSI, Milan Italy) and analyzed by the software Gen5 (BioTek, AHSI, Milan Italy).

### 2.9. Chromatin Immunoprecipitation (ChIP) Assay

The ChIP experiments were carried out, normalized and analyzed as described [28]. The primers used to amplify a region containing an AP-1 site located in the IL-8 promoter sequence were as follow: 5′-GTTCTAACACCTGCCACTCT-3′ (Fwd) and 5′-CCACGATTTGCAACTGATGG-3′ (Rev).

### 2.10. Conditioned Medium 

The CAFs were seeded in regular medium in six-well multi-dishes and the next day were switched to a medium without serum. After 24 h, the CAFs were treated for 6 h with AGEs, as indicated; thereafter, the cells were washed twice with PBS and cultured for an additional 12 h with fresh serum-free medium. The supernatants were collected, centrifuged at 3500 rpm for 5 min to remove cellular debris and used as the conditioned medium for the appropriate experiments.

### 2.11. Acetone Precipitation of Proteins

The protein precipitation from the conditioned medium derived from the CAFs was performed by using the precipitation method with acetone, as previously reported [28,70,71]. Briefly, four volumes of ice-cold acetone (Sigma-Aldrich, Milan, Italy) were added to one volume of sample, the mixture was then vortexed and incubated at -20 °C overnight. This was followed by centrifugation at 10,000× *g* for 15 min at 4 °C. Thereafter, the supernatants were discarded, the pellet was air dried, dissolved in Laemmli buffer 2× and used in the appropriate experiments. In the Western blot analysis, the protein loading of the conditioned medium samples was checked by Ponceau red staining (0.1% Ponceau S (*w*/*v*) in 5% acetic acid) of the blotted membranes.

### 2.12. Polarization Assay

The MDA-MB-231 cells were serum deprived for 24 h and then exposed for 6 h to the conditioned medium collected from the CAFs treated as indicated. Then, the cells were processed as previously described [72,73].

### 2.13. Transwell Migration and Invasion Assays

Transwell Migration and Invasion Assays were carried out, as previously described [74]. Briefly, the transwell 8 µm polycarbonate membrane (Costar, Sigma-Aldrich, Milan, Italy) was used to evaluate in vitro migration and invasion of MDA-MB-231 cells. A total of 5 × 10^4^ cells in 300 µL serum-free medium were seeded in the upper chamber, coated with (invasion assay) or without (migration assay) Corning^®^ Matrigel^®^ Growth Factor Reduced (GFR) Basement Membrane Matrix (Biogenerica, Catania, Italy), diluted with serum-free medium at a ratio of 1:3. Next, the conditioned medium from the CAFs, treated as indicated, was added to the bottom chambers in the presence or absence of 300 ng/mL Ab-IL-8 or the CXCR1/2 inhibitor reparixin, where required. On reaching 6 h after seeding, the cells on the upper surface of the membrane were removed by wiping with a Q-tip, and the invaded or migrated cells were fixed with 100% methanol and stained with Giemsa (Sigma-Aldrich, Milan, Italy). The images were acquired using the Cytation 3 Cell Imaging Multimode Reader (BioTek, AHSI, Milan Italy) and the cells were counted using the WCIF ImageJ software (National Institutes of Health (NIH), Rockville Pike, Bethesda, MD, USA).

### 2.14. Statistical Analysis

The data were analyzed by one-way ANOVA with Dunnett’s multiple comparisons where applicable, using GraphPad Prism version 6.01 (GraphPad Software, Inc., San Diego, CA, USA). The Kaplan–Meier *p*-value is based on a log-rank test. (*) *p* < 0.05 was considered statistically significant.

## 3. Results

### 3.1. AGEs Induce ERK and AKT Phosphorylation in Breast CAFs

Previous studies have shown that the ERK1/2 and AKT transduction pathways act as critical mediators involved in the AGEs-dependent responses in both normal and cancer cells [75,76,77]. In order to provide novel insights into the AGEs-mediated signaling within the tumor microenvironment, we sought to address whether AGEs may also activate ERK1/2 and AKT in the CAFs obtained from breast tumor patients. Both ERK and AKT phosphorylations were observed in a time-dependent manner upon AGEs exposure in the CAFs (Figure 1a), whereas the incubation with BSA alone had no effect (Appendix A). Next, we verified whether RAGE is involved in these stimulatory effects elicited by the AGEs in the CAFs. By pharmacological and gene silencing approaches, we found that the activation of the ERK and AKT induced by AGEs is prevented in the presence of the RAGE inhibitor, FPS-ZM1 (Figure 1b), as well as transfecting the CAFs with siRNA sequences targeting RAGE (Figure 1c; Appendix A). Reminiscing from previous data on the ability of AGEs to induce ROS levels in diverse cell contexts [21,78,79,80], we found that the AGEs treatment triggers the generation of ROS, however, this response was no longer observed either using the RAGE inhibitor FPS-ZM1 (Figure 1d) or the ROS scavenger NAC (Figure 1e). Thereafter, we established that the phosphorylation of ERK and AKT upon AGEs exposure is strictly dependent on ROS generation, as ascertained by the ROS scavenger, NAC (Figure 1f).

### 3.2. The Activation of AGEs/RAGE Transduction Pathway Up-Regulates IL-8 Levels in CAFs

A growing body of evidence pointed out that RAGE is an important mediator connecting chronic inflammation to neoplastic progression through the various autocrine and/or paracrine interactions, which occur between the cancer cells and the components of the surrounding microenvironment [23,25,81]. Hence, in order to investigate the pro-inflammatory gene expression profile elicited by the AGEs/RAGE activation within the breast tumor stroma, we performed TaqMan™ gene expression experiments by Human Chemokine Array. To this end, the CAFs were treated with AGEs in the presence or absence of the RAGE inhibitor, FPS-ZM1 (Figure 2a). Thereafter, we focused our attention on the genes displaying a CT < 32 along with at least 0.58 log2 fold change upon the AGEs exposure, in relation to the vehicle-treated CAFs. IL-8 (also known as CXCL8) was the most upregulated gene, whose expression was abrogated using the RAGE inhibitor, FPS-ZM1. The induction of IL-8 by AGEs and the repressive effects elicited by the RAGE inhibitor FPS-ZM1 were confirmed at both the mRNA and protein levels, respectively, by real-time PCR and Western blotting assays (Figure 2b,c). Similar results were then obtained by silencing the RAGE expression (Figure 2d; Appendix A). Furthermore, the AGEs-mediated increase in IL-8 was abolished at both the mRNA and protein levels, using the ROS scavenger, NAC (Figure 2e,f). Considering that the activation of the AGEs/RAGE signaling triggers the phosphorylation of ERK and AKT, we assessed whether the IL-8 upregulation is no longer evident, using the MEK inhibitor trametinib as well as the PI3K inhibitor alpelisib (Figure 2g). Next, the secretion of IL-8 upon AGEs exposure was evaluated in the conditioned medium collected from CAFs. Of note, the upregulation of the IL-8 levels prompted by AGEs in the conditioned medium derived from CAFs was prevented by the RAGE inhibitor, FPS-ZM1, and by knocking down the expression of RAGE (Figure 2h,i). In addition, we ascertained that the ROS scavenger NAC abrogates the secretion of IL-8 elicited by AGEs in the CAFs medium (Figure 2j).

### 3.3. c-Fos Is Involved in the Up-Regulation of IL-8 Prompted by AGEs/RAGE Signaling in CAFs

Based on the above data, we aimed to provide mechanistic insights into the upregulation of IL-8 promoted by the AGEs/RAGE axis in CAFs. Of note, we found that the transcriptional activation of the IL-8 promoter construct elicited by AGEs was no longer evident using the RAGE inhibitor FPS-ZM1, the ROS scavenger NAC, the MEK inhibitor trametinib and the PI3K inhibitor alpelisib (Figure 3a). Considering that the previous investigations have highlighted the main role played by c-Fos in the regulation of the IL-8 expression upon exposure to diverse treatments [82,83,84,85], we ascertained that the transactivation of the c-Fos promoter-construct upon AGEs treatment is abolished in the presence of the RAGE inhibitor FPS-ZM1, the ROS scavenger NAC, the MEK inhibitor trametinib and the PI3K inhibitor alpelisib (Figure 3b). Thereafter, performing real-time PCR and Western blotting assays, we established that the AGEs trigger c-Fos expression at both the mRNA and protein levels, however these responses were no longer evident in the presence of the RAGE inhibitor FPS-ZM1, silencing RAGE expression and using the ROS scavenger NAC (Figure 3c–g; Appendix A). Furthermore, the AGEs-mediated increase in the c-Fos expression was abrogated using the MEK inhibitor trametinib as well as the PI3K inhibitor alpelisib (Figure 3h). Intriguingly, the chromatin immunoprecipitation (ChIP) assays performed in the CAFs treated with AGEs revealed that c-Fos is recruited to the AP-1 site located within the human IL-8 promoter sequence (Figure 3i,j). Further supporting these results, the treatment with AGEs did not induce the transactivation of IL-8 promoter construct in CAFs previously transfected with a dominant negative form of c-Fos (DN/c-Fos) (Figure 3k). In accordance with these findings, the upregulation and secretion of IL-8 elicited by AGEs were prevented from transfecting the CAFs with the DN/c-Fos expression vector (Figure 3l,m).

### 3.4. IL-8/CXCR1/2 Paracrine Activation Promotes the Acquisition of a Migratory and Invasive Phenotype of MDA-MB-231 Breast Cancer Cells

Previous studies have revealed that the production and secretion of soluble factors by CAFs activate inflammatory responses, leading to the acquisition of aggressive and pro-metastatic cancer phenotypes [86,87]. In this regard, chemokines and cytokines have been shown to play a critical role in the migration and invasion of breast cancer cells [53,55,88,89]. For instance, IL-8 binding to the cognate receptors, namely CXCR1 and CXCR2, may promote a dysfunctional inflammatory microenvironment that contributes to tumor progression [84,89,90,91,92,93]. On the basis of these observations, we performed an in silico evaluation by querying the Affymetrix dataset, which supplies the clinical information and mRNA expression data of a large cohort of breast tumor patients. Of note, a higher expression of the mRNA levels of CXCR1/2 was found in basal breast cancer subtype in relation to the luminal A, luminal B, ERBB2 and normal-like molecular subgroups (Figure 4a). In addition, the survival analysis performed on the basal breast cancer subtype revealed that a worse relapse free survival (RFS) characterizes patients exhibiting a high expression of CXCR1/2 (Figure 4b; Appendix A). Next, we aimed to evaluate whether IL-8 secreted by CAFs may promote a feed-forward loop that engages CXCR1/2 toward the acquisition of certain malignant features in the MDA-MB-231 breast cancer cells, which express elevated CXCR1/2 levels [94,95]. Worthy, conditioned-medium collected from the AGEs-stimulated CAFs triggered a spindle-like morphology in the MDA-MB-231 cells, which was prevented using the RAGE inhibitor FPS-ZM1 (Figure 4c). The abovementioned effects were also obtained by the neutralizing antibody anti-IL-8 (Figure 4b), as well as the CXCR1/2 inhibitor reparixin (Figure 4c).

Accordingly, the RAGE inhibitor FPS-ZM1 prevented the formation of stress fibers in the MDA-MB-231 cells exposed to the conditioned medium from the AGEs-stimulated CAFs, as assessed through immunofluorescent staining of the polymerized actin (F-actin) (Figure 5a). Similar results were obtained in the presence of the neutralizing IL-8 antibody (Figure 5b), as well as the CXCR1/2 inhibitor reparixin (Figure 5c). To further appreciate the biological significance of the paracrine action of IL-8 in breast malignancy, we demonstrated that the RAGE inhibitor FPS-ZM1 (Figure 6a,b), the neutralizing anti-IL-8 antibody (Figure 6c,d) and the CXCR1/2 inhibitor reparixin (Figure 6e,f) abrogate the migration and invasion of the MDA-MB-231 cells cultured in the conditioned medium from the AGEs-treated CAFs. Next, the mRNA expression of certain Epithelial-to-Mesenchymal Transition (EMT) biomarkers, namely N-cadherin, vimentin and fibronectin, increased in the MDA-MB-231 cells exposed to the conditioned medium collected from AGEs-stimulated CAFs. However, these responses were no longer evident in the presence of the RAGE inhibitor FPS-ZM1 (Appendix A).

## 4. Discussion

In the current study, we investigated the role of the AGEs/RAGE transduction pathway in the multifaceted interaction that occurs between breast cancer cells and the tumor microenvironment. In this vein, we first assessed that the activation of the AGEs/RAGE axis induces rapid ERK and AKT phosphorylation via ROS generation in CAFs obtained from breast cancer patients. Performing a PCR human chemokine array we then found that IL-8 is the most upregulated gene in AGEs-stimulated CAFs. Mechanistically, we ascertained that the AGEs/RAGE signaling cascade triggers IL-8 expression through the induction of c-Fos, which was recruited within the IL-8 promoter sequence. Thereafter, we assessed that IL-8, secreted in the conditioned medium collected from the AGEs-stimulated CAFs, acts in a paracrine manner through the cognate receptors CXCR1/2, empowering the acquisition of a spindle-like morphology and the actin polymerization in MDA-MB-231 breast cancer cells. Consequently, these cells acquire migratory and invasive properties, in accordance with previous studies highlighting the role of IL-8 in stimulating pro-metastatic effects in diverse types of tumors, including breast cancer [30,31,32,33,34,35].

An intricate signaling network coordinates the regulation and secretion of IL-8 in both normal and tumor cells [96]. Cytokines, such as tumor necrosis factor-alpha (TNFα), IL-6 and IL-1β, growth factors and hormones, were reported to upregulate the IL-8 expression [30,97,98]. The key components of the tumor stroma, such as CAFs, have been also shown to promote the resistance to therapeutics and the acquisition of pro-metastatic phenotypes ensuing the production and release of IL-8 within the tumor milieu [99,100,101]. In accordance with these findings, in the present study the AGEs, through the cognate receptor RAGE and the subsequent activation of the ROS-ERK1/2-AKT-c-Fos transduction pathway, triggered the expression of IL-8 and its release by the CAFs toward the acquisition of malignant features in the breast cancer cells.

Impaired glucose tolerance and insulin resistance may increase the risk of breast cancer [102,103,104]. For instance, diabetes in premenopausal women has been correlated with a major risk of developing a breast tumor, mostly the aggressive subtype named triple-negative breast cancer (TNBC) [105,106]. In addition, previous studies have indicated that breast cancer mortality associated with distant metastasis is elevated in those patients with co-occurring low-grade chronic inflammatory conditions, such as obesity and T2DM [7,107,108,109,110]. Of note, the IL-8 serum levels were found markedly increased in the T2DM patients in relation to the healthy subjects, and positively correlated with worse metabolic control and an inflammatory state [111]. In this regard, it is worth mentioning that IL-8 levels were found elevated in serum of cancer patients and correlated with a high tumor grade and an unfavorable clinical outcome [33,43,70]. Furthermore, elevated levels of IL-8 were involved in the resistance to treatments in diverse types of cancers like pancreatic, colorectal and breast tumors [112,113,114]. Likewise, pharmacological or genetic inhibition of IL-8 were shown to sensitize breast cancer cells to the cytotoxic effects of conventional chemotherapy agents [115]. The biological effects of IL-8 are mediated via two rhodopsin-like G-protein-coupled receptors (GPCRs): CXCR1 and CXCR2, also known, respectively, as IL-8RA and IL-8RB [116,117]. Similar to other GPCRs, CXCR1 and CXCR2 are composed of seven transmembrane domains, with extracellular N-terminus and intracellular C-terminus portions [118,119]. The IL-8/CXCR1/2 axis has been involved in the stimulation of diverse signaling pathways, including PI3K, MAPK, JAK/STAT and RhoGTPase, which in turn mediate biological responses, such as proliferation, survival, invasion, cytoskeletal dynamics and angiogenesis [40,120]. Moreover, the IL-8/CXCR1/2 system correlates with poor clinical prognosis in diverse types of tumor, including breast cancer [121,122,123,124,125]. Moreover, a significant upregulation of the CXCR1/2 levels were detected in the invasive breast carcinoma samples in relation to normal breast tissues [89]. Nicely fitting with these data, the blockade of IL-8 by a neutralizing antibody or the inhibition of CXCR1/2, blunted tumor growth and metastasis, and also reversed the resistance to treatments in breast cancer [89,91,94,126]. In accordance with the aforementioned findings, our in silico analysis of the Affymetrix dataset displayed higher expression levels of CXCR1/2 in the basal subtype in relation to the other molecular subgroups of breast tumors. Likewise, the evaluation of the Kaplan–Meier survival curves revealed a worse clinical outcome in basal breast cancer patients showing high CXCR1/2 levels. Providing further insights on the role of the IL-8/CXCR1/2 axis within the tumor microenvironment, we also assessed that the paracrine actions elicited by the conditioned medium collected from the AGEs-stimulated CAFs are no longer evidently inhibiting this axis.

## 5. Conclusions

Our findings assessed for the first time the role of the AGEs/RAGE signaling pathway in breast CAFs and its involvement in the paracrine stimulation of the IL-8/CXCR1/2 axis toward the acquisition of malignant features of breast cancer cells. However, further studies are needed to better characterize the transduction cascade triggered by the IL-8/CXCR1/2 paracrine activation in the MDA-MB-231 cells. Likewise, in vivo studies are warranted in order to corroborate the present results and the usefulness of IL-8 as a therapeutic target in comprehensive approaches halting breast cancer progression.

## Figures and Tables

**Figure 1 cells-11-02402-f001:**
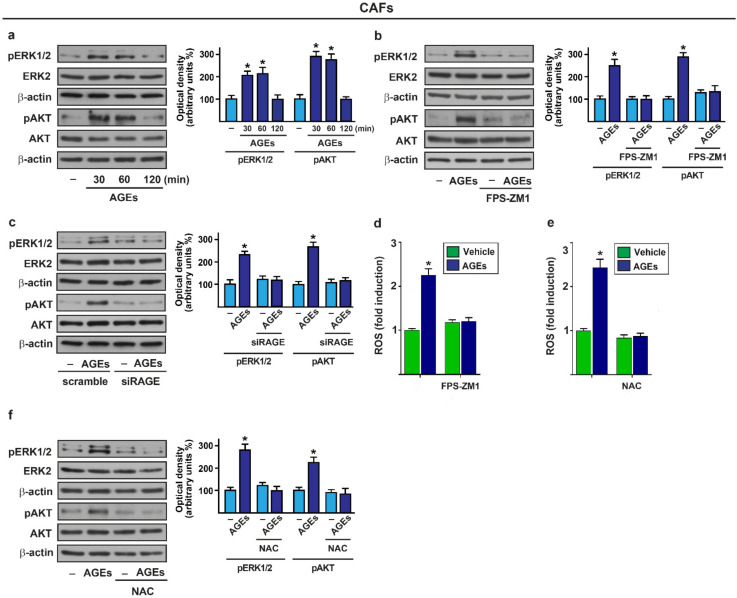
AGEs trigger rapid responses via RAGE in CAFs. (**a**) Phosphorylation of ERK1/2 and AKT in CAFs exposed to vehicle (–) and 100 µg/mL AGEs for the indicated times; (**b**) Phosphorylation of ERK1/2 and AKT in CAFs treated for 30 min with vehicle (–) or 100 µg/mL AGEs alone and in the presence of 1 μM RAGE inhibitor FPS-ZM1; (**c**) Immunoblots of ERK1/2 and AKT in CAFs transfected with scramble siRNA or siRAGE (10 nM) for 24 h and then treated for 30 min with vehicle (–) and 100 µg/mL AGEs; (**d**,**e**) ROS generation in CAFs exposed to vehicle and 100 µg/mL AGEs alone or in the presence of 1 μM RAGE inhibitor FPS-ZM1 and 300 µM free radical scavenger N-acetyl-Lcysteine (NAC), as indicated. The values of fluorescent probe DCF-DA obtained in CAFs treated with vehicle was set as one-fold induction upon which ROS levels induced by AGEs were calculated. Values represent the mean ± SD of three independent experiments performed in triplicate; (**f**) The activation of ERK1/2 and AKT in CAFs upon 100 µg/mL AGEs exposure for 30 min was abolished using 300 µM free radical scavenger NAC. ERK2 and AKT were used as loading control, as indicated. Side panels show densitometric analysis of the blots normalized to the loading controls. Values represent the mean ± SD of three independent experiments. (*) indicates *p* < 0.05.

**Figure 2 cells-11-02402-f002:**
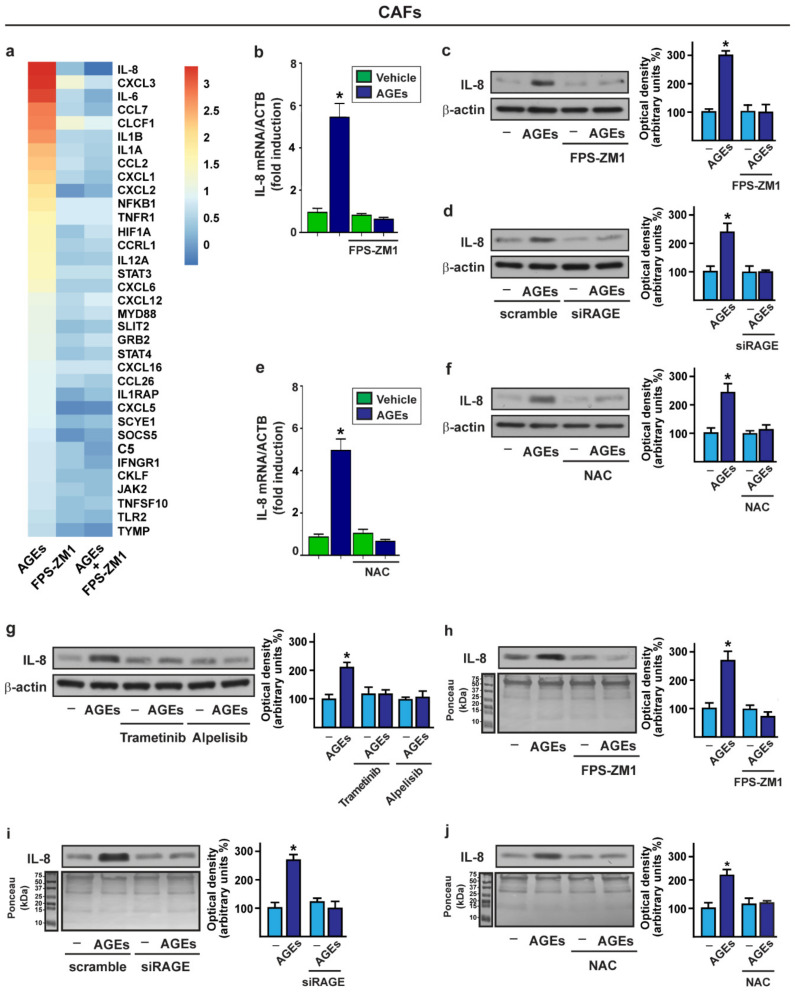
AGEs/RAGE activation upregulates IL-8 expression in CAFs. (**a**) CAFs were exposed for 8 h to 100 µg/mL AGEs alone and in the presence of 1 μM RAGE inhibitor FPS-ZM1. Gene expression changes of chemokine and related genes were evaluated by TaqMan™ Human Chemokine Array. Values were normalized to 18 S expression; the colors indicate the log2 fold changes of gene expression upon the indicated conditions in relation to the vehicle-treated CAFs. mRNA (**b**,**e**) and protein (**c**,**f**) expression of IL-8 evaluated, respectively, by real-time PCR and immunoblotting in CAFs treated for 8 h with vehicle (–) or 100 µg/mL AGEs alone and in the presence of 1 μM RAGE inhibitor FPS-ZM1 or in combination with 300 µM free radical scavenger NAC. In RNA experiments, values were normalized to the beta-actin (ACTB) expression and shown as fold changes of IL-8 mRNA expression upon AGEs treatment compared to cells exposed to vehicle; (**d**) Immunoblots showing IL-8 protein expression in CAFs transfected with scramble siRNA or siRAGE (10 nM) for 24 h and then exposed for 8 h to vehicle (–) or 100 µg/mL AGEs; (**g**) IL-8 protein expression evaluated by immunoblotting in CAFs treated for 8 h with vehicle (–) or 100 µg/mL AGEs alone and in combination with 100 nM MEK inhibitor trametinib or 1 µM PI3K inhibitor alpelisib. β-actin served as a loading control; (**h**,**j**) Evaluation by immunoblotting of IL-8 protein levels in conditioned medium (CM) collected from CAFs treated for 18 h with vehicle (–) or 100 µg/mL AGEs alone and in the presence of 1 μM RAGE inhibitor FPS-ZM1 or in combination with 300 µM free radical scavenger NAC; (**i**) Immunoblots showing IL-8 protein levels in CM derived from CAFs transfected with scramble siRNA or siRAGE (10 nM) for 24 h and then exposed for 18 h to vehicle (–) or 100 µg/mL AGEs. Ponceau red staining of the membrane was used as a loading control for the CM. Side panels show densitometric analysis of the blots normalized to the loading controls. Values represent the mean ± SD of three independent experiments. (*) indicates *p* < 0.05.

**Figure 3 cells-11-02402-f003:**
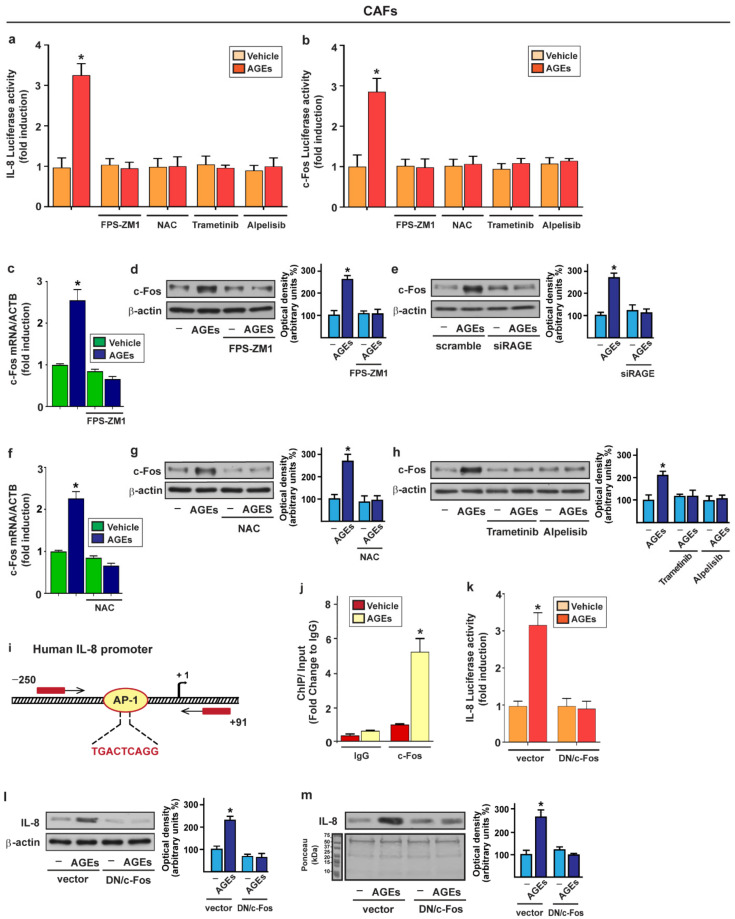
c-Fos is involved in the upregulation of IL-8 induced by AGEs/RAGE signaling in CAFs. (**a**) Luciferase activities of IL-8 promoter construct in CAFs treated for 18 h with vehicle or 100 µg/mL AGEs in the presence or absence of 1 μM RAGE inhibitor FPS-ZM1, 300 µM free radical scavenger NAC, 100 nM MEK inhibitor trametinib or 1 µM PI3K inhibitor alpelisib, as indicated; (**b**) Luciferase activities of c-Fos promoter construct in CAFs upon exposure for 18 h to vehicle or 100 µg/mL AGEs alone and in combination with 1 μM RAGE inhibitor FPS-ZM1, 300 µM free radical scavenger NAC, 100 nM MEK inhibitor trametinib or 1 µM PI3K inhibitor alpelisib, as indicated. The luciferase activities were normalized to the internal transfection control, and values of cells receiving vehicle were set as 1-fold induction upon which the activity induced by AGEs was calculated. Each column represents the mean ± SD of three independent experiments performed in triplicate. mRNA (**c**,**f**) and protein (**d**,**g**) expression of c-Fos evaluated, respectively, by real-time PCR and immunoblotting in CAFs treated for 4 h with vehicle (–) or 100 µg/mL AGEs alone and in the presence of 1 μM RAGE inhibitor FPS-ZM1 or in combination with 300 µM free radical scavenger NAC. In RNA experiments, values were normalized to the beta-actin (ACTB) expression and shown as fold changes of c-Fos mRNA expression upon AGEs treatment compared to cells exposed to vehicle; (**e**) Immunoblots showing c-Fos protein expression in CAFs transfected with non-targeting scramble siRNA or siRAGE (10 nM) for 24 h and then exposed for 4 h with vehicle (–) or 100 µg/mL AGEs; (**h**) c-Fos protein expression evaluated by immunoblotting in CAFs treated for 4 h with vehicle (–) or 100 µg/mL AGEs alone and in combination with 100 nM MEK inhibitor trametinib or 1 µM PI3K inhibitor alpelisib; (**i**,**j**) Recruitment of c-Fos to the AP-1 site located within the IL-8 promoter region upon treatment for 4 h with AGEs in CAFs, as assessed by Chromatin Immunoprecipitation (ChIP) assays. Data obtained were normalized to the input and shown as fold changes in relation to nonspecific Immunoglobulin G (IgG). Each column represents the mean ± SD of three independent experiments performed in triplicate; (**k**) Luciferase activities of IL-8 promoter construct in CAFs transfected for 18 h with an empty vector or a plasmid encoding for a dominant negative form of c-Fos (DN/c-Fos) and then exposed for 18 h to vehicle or 100 µg/mL AGEs. The luciferase activities were normalized to the internal transfection control and values of cells receiving vehicle were set as 1-fold induction upon which the activity induced by treatment was calculated. Each column represents the mean ± SD of three independent experiments performed in triplicate; (**l**) IL-8 protein expression evaluated by immunoblotting in CAFs transfected with the empty vector or with the DN/c-Fos construct for 18 h and then treated with vehicle (–) or 100 µg/mL AGEs for 8 h. β-actin served as a loading control; (**m**) Immunoblotting of IL-8 in conditioned medium (CM) collected from CAFs transfected with a vector or with the DN/c-Fos construct and then treated for 8 h with vehicle (–) or 100 µg/mL AGEs. Ponceau red staining of the membrane was used as a loading control for the CM. Side panels show densitometric analysis of the blots normalized to the loading controls. Data shown represent the mean ± SD of three independent experiments. (*) indicates *p* < 0.05.

**Figure 4 cells-11-02402-f004:**
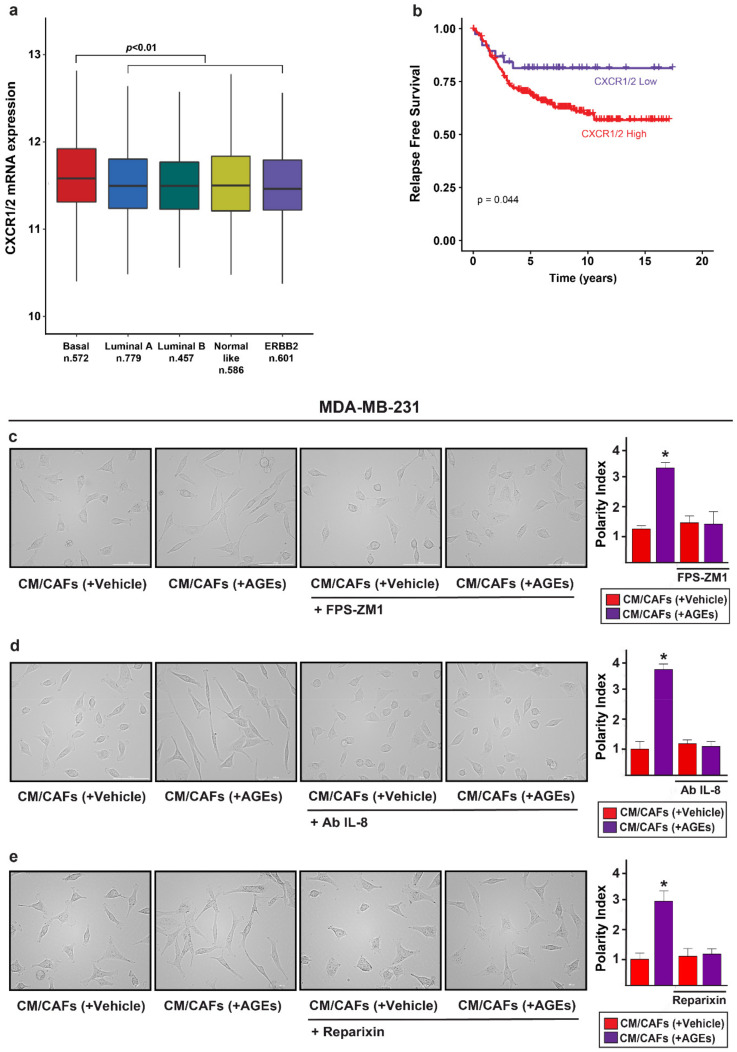
IL-8 mediates the acquisition of a spindle-like morphology in MDA-MB-231 cells triggered by conditioned medium (CM) from AGEs-stimulated CAFs. (**a**) CXCR1/2 mRNA levels according to breast cancer intrinsic molecular subtypes of the integrated Affymetrix cohort; (**b**) CXCR1/2 expression is associated with a worse relapse-free survival (RFS) of basal breast cancer patients in the Affymetrix dataset. The patients were divided into high and low CXCR1/2 expression levels on the basis of the established cut-point’ (**c**) MDA-MB-231 cells were incubated for 6 h with CM collected from CAFs previously treated with vehicle or 100 µg/mL AGEs in the presence or absence of 1 μM RAGE inhibitor FPS-ZM1’ (**d**) MDA-MB-231 cells were cultured for 6 h in CM derived from CAFs previously treated with vehicle or 100 µg/mL AGEs with or without 300 ng/mL IL-8 neutralizing-antibody (Ab IL-8) as well as (**e**) using 5 μM CXCR1/2 inhibitor reparixin. The spindle-like morphology was quantified as Polarity Index (PI). PI = 1.0 indicates a polygonal shape, conversely a value > 1.0 identified ranges of migratory shapes. Images shown are representative of 10 random fields acquired in three independent experiments. Scale bar = 100 µm. (*) indicates *p* < 0.05.

**Figure 5 cells-11-02402-f005:**
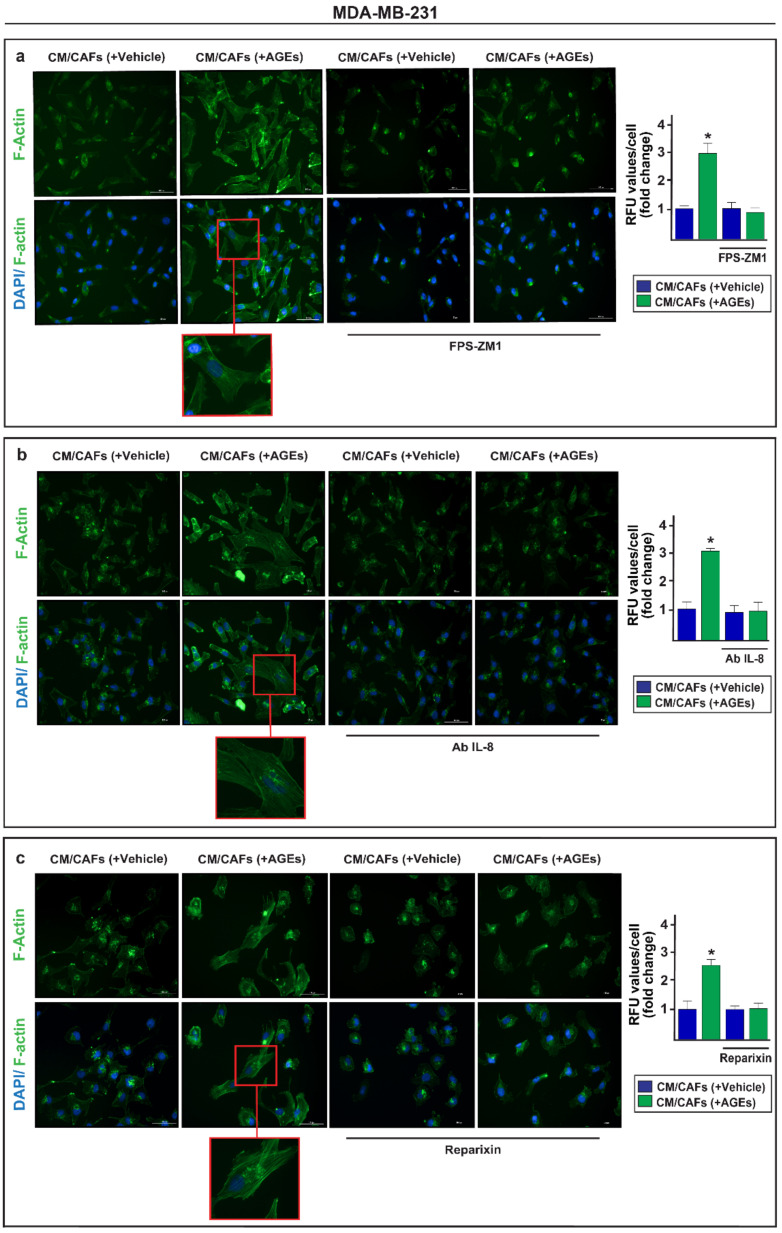
The paracrine action of IL-8 promotes the formation of actin stress fibers in MDA-MB-231 cells. (**a**) MDA-MB-231 cells, which were treated for 6 h with conditioned medium (CM) from CAFs exposed to vehicle or 100 µg/mL AGEs alone and in the presence of 1 μM RAGE inhibitor FPS-ZM1, were stained with FITC-conjugated phalloidin to visualize F-actin stress fibers (green) and DAPI to detect nuclei (blue). The F-actin stress fibers formation in MDA-MB-231 cells promoted by CM collected from CAFs previously treated with 100 µg/mL AGEs, was abrogated using 300 ng/mL IL-8 neutralizing-antibody (Ab IL-8) (**b**) or 5 μM CXCR1/2 inhibitor reparixin (**c**). Fluorescence intensities of the number of fibers/cell was quantified by F-actin staining in 10 random fields for each condition; results are expressed as fold change of relative fluorescence units (RFU). Data shown represent the mean ± SD of three independent experiments performed in triplicate. (*) indicates *p*< 0.05. Enlarged details are shown in the separate boxes. Scale bar 100 μM.

**Figure 6 cells-11-02402-f006:**
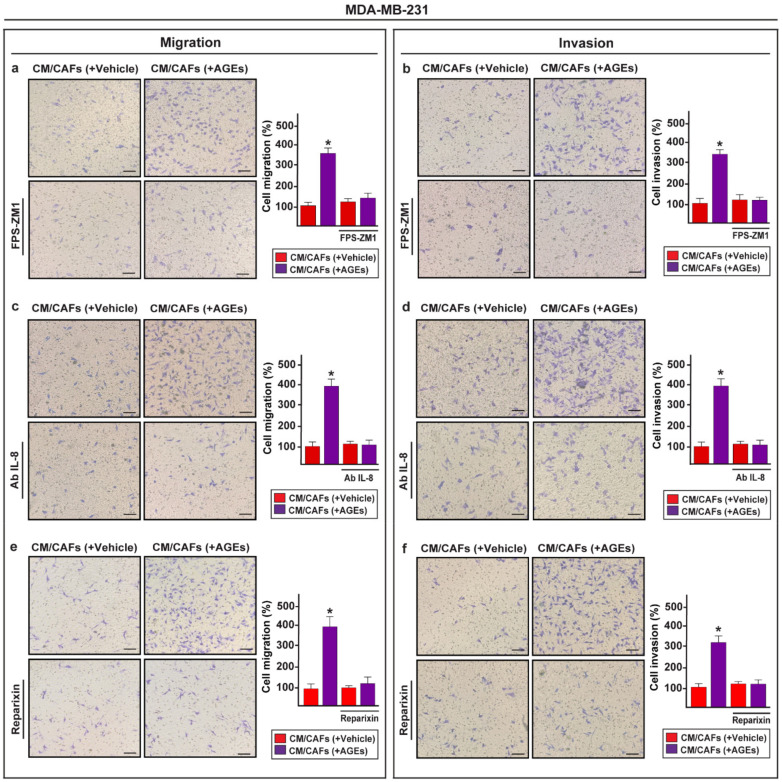
The paracrine activation of IL-8/CXCR1/2 axis promotes cell migration and invasion of MDA-MB-231 cells. Transwell assays were performed to evaluate cell migration (**a**) and invasion (**b**) in MDA-MB-231 cells cultured for 6 h in conditioned medium (CM) from CAFs previously treated with vehicle or 100 µg/mL AGEs alone and in combination with 1 μM RAGE inhibitor FPS-ZM1. Cell migration (**c**) and invasion (**d**) were assessed in MDA-MB-231 cells cultured for 6 h in conditioned medium (CM) from CAFs previously treated with vehicle or 100 µg/mL AGEs alone and in combination with 300 ng/mL IL-8 neutralizing-antibody (Ab IL-8). The migration (**e**) and invasion (**f**) of MDA-MB-231 cells observed upon exposure to CM from CAFs previously treated with 100 µg/mL AGEs were abolished using 5 μM CXCR1/2 inhibitor reparixin. Scale bar = 200 µm. Side panels show the mean ± SD of the number of cells counted in at least 10 random fields of three independent experiments performed in triplicate. (*) indicates *p* < 0.05.

## Data Availability

Publicly available datasets were analyzed and data can be found at: https://www.ncbi.nlm.nih.gov/geo/ (accessed on 1 July 2022). (Accession numbers: GSE12276, GSE21653, GSE3744, GSE5460, GSE2109, GSE1561, GSE17907, GSE2990, GSE7390, GSE11121, GSE16716, GSE2034, GSE1456, GSE6532, GSE3494).

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
