# Peer review of "The AGEs/RAGE Transduction Signaling Prompts IL-8/CXCR1/2-Mediated Interaction between Cancer-Associated Fibroblasts (CAFs) and Breast Cancer Cells"

_cells, 2022, doi:10.3390/cells11152402_

Round 1

Reviewer 1 Report

Dear authors, I have reviewed your excellent manuscript entitled: ‘The AGEs/RAGE transduction signalling prompts IL-8/CXCR1/2-mediated interaction between cancer-associated fibroblasts (CAFs) and breast cancer cells ’ and I have the following comments:

Major points:
1. Introduction should be extended by information related to biological role of IL-8 in cancer.
2. Definitely authors should avoid discussion in results section. In results section please describe only your outcomes.
3. In paragraph 2.14 you did not describe all statistical test. How you got p-value in Kaplan-Meier plot? What kind of statistical test you have applied?
4. How you estimated cut-off point for low and high CXCR1/1 in plot 4b?
5. The second paragraph of discussion is a part of introduction.
6. Authors should provide limitation of the study.

Minor points:
1) Please improve the quality of figure 5.
2) Also, the symbols could be little bit bigger in all figures.

A curiosity: How the study results can be translated into clinical practice?

Author Response

Dear authors, I have reviewed your excellent manuscript entitled: ‘The AGEs/RAGE transduction signalling prompts IL-8/CXCR1/2-mediated interaction between cancer-associated fibroblasts (CAFs) and breast cancer cells’ and I have the following comments:

Response: We are grateful to the Reviewer for appreciating our work.

Major points:

  1. Introduction should be extended by information related to biological role of IL-8 in cancer.

Response: In the revised manuscript we have extended the introduction providing information regarding the biological role of IL-8 in cancer.

  1. Definitely authors should avoid discussion in results section. In results section please describe only your outcomes.

Response: In the revised manuscript we have removed discussion in results section.

  1. In paragraph 2.14 you did not describe all statistical test. How you got p-value in Kaplan-Meier plot? What kind of statistical test you have applied?

Response: The log-rank test was the statistical test applied in the Kaplan-Meier plot, we have specified this point in the revised version of the manuscript.

  1. How you estimated cut-off point for low and high CXCR1/1 in plot 4b?

Response: The cut-off point was calculated using the survivALL() function of the survivALL R package (Pearce et al.; doi:10.1101/208660). The function calculates the Cox proportional hazards for all possible points‐of‐separation and validates the point-of-separation which most clearly distinguishes (lowest p-value) between good and poor prognosis. An additional panel representing this analysis has been added to the supplementary material of the revised manuscript.

  1. The second paragraph of discussion is a part of introduction.

Response: In the revised manuscript the second paragraph of discussion has been included in the introduction.

  1. Authors should provide limitation of the study.

Response: In the revised manuscript we have provided some limitations of the study.

Minor points:

  • Please improve the quality of figure 5.

Response: The quality of figure 5 has been improved.

  • Also, the symbols could be little bit bigger in all figures.

Response: The symbols of the figures are bigger in the revised manuscript.

A curiosity: How the study results can be translated into clinical practice?

Response: The present study holds potential from a translational standpoint. Approximately 20% of breast cancer patients is affected by metabolic disorders like obesity, hyperglycaemia and type-2 diabetes. Currently, no specific guidelines are available for the treatment of breast cancer patients affected by the aforementioned diseases. Consequently, these patients lack effective therapeutic treatments and unfortunately elevated mortality rates. Therefore, our study may pave the way for setting a combination therapy targeting both RAGE and CXCR1/2 for a better therapeutic management of breast cancer patients affected by metabolic disorders.

Reviewer 2 Report

Santolla and collogues demonstrated that, upon AGEs/RAGE activation, primary breast cancer-associated fibroblasts increase IL-8 secretion. Interestingly, they demonstrate that IL-8/CXCR1/2 paracrine activation promotes a migratory and invasive phenotype in breast cancer cells.

The manuscript is, in general, clear and presented in a well-structured manner. The topic is interesting, while presented results are not of great novelty in the field of breast cancer research. The experimental design is appropriate to test the hypothesis, and the study is correctly designed and technically accurate. The authors adopted different strategies and presented data are robust enough to demonstrate their conclusion to draw conclusions.

The paper can, in principle, be accepted after revision based on the following suggestions:

-Authors correctly performed transwell migration and invasion assays to demonstrate that the paracrine activation of IL-8/CXCR1/2 promotes cell motility. However, to have a more precise description of the role of this axis on the epithelial-mesenchymal transition, authors are suggested to evaluate the expression of the mRNAs encoding E-cadherin, N-cadherin, vimentin, and fibronectin in MDA-MB-231 cells following the incubation with CM from AGEs-stimulated CAFs with or without the stimulation with FPS-ZM1 or anti-IL-8 antibody.

-Recently, it has been established that IL-8, via its cognate receptors CXCR1 and CXCR2, is also involved in regulating breast cancer stem-like cell activity. Therefore, authors are suggested to evaluate mammosphere formation and CD44high/CD24low antigenic phenotype in MDA-MB-231 cells following the incubation with CM from AGEs-stimulated CAFs to evaluate the role of the identified CAF-induced IL-8/CXCR1/2 axes on the cancer stem-like phenotype.  

Author Response

Santolla and collogues demonstrated that, upon AGEs/RAGE activation, primary breast cancer-associated fibroblasts increase IL-8 secretion. Interestingly, they demonstrate that IL-8/CXCR1/2 paracrine activation promotes a migratory and invasive phenotype in breast cancer cells.

The manuscript is, in general, clear and presented in a well-structured manner. The topic is interesting, while presented results are not of great novelty in the field of breast cancer research. The experimental design is appropriate to test the hypothesis, and the study is correctly designed and technically accurate. The authors adopted different strategies and presented data are robust enough to demonstrate their conclusion to draw conclusions.

Response: We are grateful to the Reviewer for the encouraging remarks.

The paper can, in principle, be accepted after revision based on the following suggestions:

-Authors correctly performed transwell migration and invasion assays to demonstrate that the paracrine activation of IL-8/CXCR1/2 promotes cell motility. However, to have a more precise description of the role of this axis on the epithelial-mesenchymal transition, authors are suggested to evaluate the expression of the mRNAs encoding E-cadherin, N-cadherin, vimentin, and fibronectin in MDA-MB-231 cells following the incubation with CM from AGEs-stimulated CAFs with or without the stimulation with FPS-ZM1 or anti-IL-8 antibody.

Response: As suggested, we have performed real-time PCR assays in order to evaluate the expression of N-cadherin, vimentin, and fibronectin in MDA-MB-231 cells upon exposure to CM from AGEs-stimulated CAFs in the presence/absence of the RAGE inhibitor FPS-ZM1 (Figure S3 of the revised manuscript). As it concerns E-cadherin, its expression was not detectable in MDA-MB-231 cells in accordance with previous studies showing that E-cadherin expression is suppressed by promoter methylation in these cells (doi: 10.1002/cam4.347, doi: 10.1186/1476-4598-9-179).

-Recently, it has been established that IL-8, via its cognate receptors CXCR1 and CXCR2, is also involved in regulating breast cancer stem-like cell activity. Therefore, authors are suggested to evaluate mammosphere formation and CD44high/CD24low antigenic phenotype in MDA-MB-231 cells following the incubation with CM from AGEs-stimulated CAFs to evaluate the role of the identified CAF-induced IL-8/CXCR1/2 axes on the cancer stem-like phenotype. 

Response: We thank the Reviewer for the suggestion. In next studies we will investigate the mammosphere formation and CD44high/CD24low antigenic phenotype in MDA-MB-231 cells upon exposure to CM from AGEs-stimulated CAFs, considering that the limited time for the revision does not allow us to perform these experiments requiring appropriate settings and times.